# Towards Understanding Learning Representations: To What Extent Do Different Neural Networks Learn the Same Representation

**Liwei Wang**[1,2]    **Lunjia Hu**[3]    **Jiayuan Gu**[1]    **Yue Wu**[1]
**Zhiqiang Hu**[1]    **Kun He**[4]    **John Hopcroft**[5]
[1]Key Laboratory of Machine Perception, MOE, School of EECS, Peking University
[2]Center for Data Science, Peking University, Beijing Institute of Big Data Research
[3]Computer Science Department, Stanford University
[4]Huazhong University of Science and Technology
[5]Cornell University
wanglw@cis.pku.edu.cn    lunjia@stanford.edu
{gujiayuan, frankwu, huzq}@pku.edu.cn
brooklet60@hust.edu.cn, jeh17@cornell.edu

## Abstract

It is widely believed that learning good representations is one of the main reasons for the success of deep neural networks. Although highly intuitive, there is a lack of theory and systematic approach quantitatively characterizing what representations do deep neural networks learn. In this work, we move a tiny step towards a theory and better understanding of the representations. Specifically, we study a simpler problem: How similar are the representations learned by two networks with identical architecture but trained from different initializations. We develop a rigorous theory based on the neuron activation subspace match model. The theory gives a complete characterization of the structure of neuron activation subspace matches, where the core concepts are maximum match and simple match which describe the overall and the finest similarity between sets of neurons in two networks respectively. We also propose efficient algorithms to find the maximum match and simple matches. Finally, we conduct extensive experiments using our algorithms. Experimental results suggest that, surprisingly, representations learned by the same convolutional layers of networks trained from different initializations are not as similar as prevalently expected, at least in terms of subspace match.

## 1   Introduction

It is widely believed that learning good representations is one of the main reasons for the success of deep neural networks [Krizhevsky et al., 2012, He et al., 2016]. Taking CNN as an example, filters, shared weights, pooling and composition of layers are all designed to learn good representations of images. Although highly intuitive, it is still illusive what representations do deep neural networks learn.

In this work, we move a tiny step towards a theory and a systematic approach that characterize the representations learned by deep nets. In particular, we consider a simpler problem: How similar are the representations learned by two networks with identical architecture but trained from different initializations. It is observed that training the same neural network from different random initializations frequently yields similar performance [Dauphin et al., 2014]. A natural question arises: do the differently-initialized networks learn similar representations as well, or do they learn totally

distinct representations, for example describing the same object from different views? Moreover, what is the granularity of similarity: do the representations exhibit similarity in a local manner, i.e. a single neuron is similar to a single neuron in another network, or in a distributed manner, i.e. neurons aggregate to clusters that collectively exhibit similarity? The questions are central to the understanding of the representations learned by deep neural networks, and may shed light on the long-standing debate about whether network representations are local or distributed.

Li et al. [2016] studied these questions from an empirical perspective. Their approach breaks down the concept of similarity into one-to-one mappings, one-to-many mappings and many-to-many mappings, and probes each kind of mappings by ad-hoc techniques. Specifically, they applied linear correlation and mutual information analysis to study one-to-one mappings, and found that some core representations are shared by differently-initialized networks, but some rare ones are not; they applied a sparse weighted LASSO model to study one-to-many mappings and found that the whole correspondence can be decoupled to a series of correspondences between smaller neuron clusters; and finally they applied a spectral clustering algorithm to find many-to-many mappings.

Although Li et al. [2016] provide interesting insights, their approach is somewhat heuristic, especially for one-to-many mappings and many-to-many mappings. We argue that a systematic investigation may deliver a much more thorough comprehension. To this end, we develop a rigorous theory to study the questions. We begin by modeling the similarity between neurons as the matches of subspaces spanned by activation vectors of neurons. The activation vector [Raghu et al., 2017] shows the neuron's responses over a finite set of inputs, acting as the representation of a single neuron.[1] Compared with other possible representations such as the weight vector, the activation vector characterizes the essence of the neuron as an input-output function, and takes into consideration the input distribution. Further, the representation of a neuron cluster is represented by the subspace spanned by activation vectors of neurons in the cluster. The subspace representations derive from the fact that activations of neurons are followed by affine transformations; two neuron clusters whose activations differ up to an affine transformation are essentially learning the same representations.

In order to develop a thorough understanding of the similarity between clusters of neurons, we give a complete characterization of the structure of the neuron activation subspace matches. We show the unique existence of the maximum match, and we prove the Decomposition Theorem: every match can be decomposed as the union of a set of simple matches, where simple matches are those which cannot be decomposed any more. The maximum match characterizes the whole similarity, while simple matches represent minimal units of similarity, collectively giving a complete characterization. Furthermore, we investigate how to characterize these simple matches so that we can develop efficient algorithms for finding them.

Finally, we conduct extensive experiments using our algorithms. We analyze the size of the maximum match and the distribution of sizes of simple matches. It turns out, contrary to prevalently expected, representations learned by almost all convolutional layers exhibit very low similarity in terms of matches. We argue that this observation reflects the current understanding of learning representation is limited.

Our contributions are summarized as follows.

1. We develop a theory based on the neuron activation subspace match model to study the similarity between representations learned by two networks with identical architecture but trained from different initializations. We give a complete analysis for the structure of matches.
2. We propose efficient algorithms for finding the maximum match and the simple matches, which are the central concepts in our theory.
3. Experimental results demonstrate that representations learned by most convolutional layers exhibit low similarity in terms of subspace match.

The rest of the paper is organized as follows. In Section 2 we formally describe the neuron activation subspace match model. Section 3 will present our theory of neuron activation subspace match. Based on the theory, we propose algorithms in Section 4. In Section 5 we will show experimental results and make analysis. Finally, Section 6 concludes. Due to the limited space, all proofs are given in the supplementary.

## 2 Preliminaries

In this section, we will formally describe the neuron activation subspace match model that will be analyzed throughout this paper. Let $\mathcal{X}$ and $\mathcal{Y}$ be the set of neurons in the same layer[2] of two networks with identical architecture but trained from different initializations. Suppose the networks are given $d$ input data $a_1, a_2, \cdots, a_d$. For $\forall v \in \mathcal{X} \cup \mathcal{Y}$, let the output of neuron $v$ over $a_i$ be $z_v(a_i)$. The representation of a neuron $v$ is measured by the *activation vector* [Raghu et al., 2017] of the neuron $v$ over the $d$ inputs, $\mathbf{z}_v := (z_v(a_1), z_v(a_2), \cdots, z_v(a_d))$. For any subset $X \subseteq \mathcal{X}$, we denote the vector set $\{\mathbf{z}_x : x \in X\}$ by $\mathbf{z}_X$ for short. The representation of a subset of neurons $X \subseteq \mathcal{X}$ is measured by the subspace spanned by the activation vectors of the neurons therein, $\mathrm{span}(\mathbf{z}_X) := \{\sum_{\mathbf{z}_x \in \mathbf{z}_X} \lambda_{\mathbf{z}_x} \mathbf{z}_x : \forall \lambda_{\mathbf{z}_x} \in \mathbb{R}\}$. Similarly for $Y \subseteq \mathcal{Y}$. In particular, the representation of an empty subset is $\mathrm{span}(\emptyset) := \{\mathbf{0}\}$, where $\mathbf{0}$ is the zero vector in $\mathbb{R}^d$.

The reason why we adopt the neuron activation subspace as the representation of a subset of neurons is that activations of neurons are followed by affine transformations. For any neuron $\tilde{x}$ in the following layer of $\mathcal{X}$, we have $z_{\tilde{x}}(a_i) = \mathrm{ReLU}(\sum_{x \in \mathcal{X}} w_x z_x(a_i) + b)$, where $\{w_x : x \in \mathcal{X}\}$ and $b$ are the parameters. Similarly for neuron $\tilde{y}$ in the following layer of $\mathcal{Y}$. If $\mathrm{span}(\mathbf{z}_X) = \mathrm{span}(\mathbf{z}_Y)$, for any $\{w_x : x \in X\}$ there exists $\{w_y : y \in Y\}$ such that $\forall a_i, \sum_{x \in X} w_x z_x(a_i) = \sum_{y \in Y} w_y z_y(a_i)$, and vice versa. Essentially $\tilde{x}$ and $\tilde{y}$ receive the same information from either $X$ or $Y$.

We now give the formal definition of a match.

**Definition 1** ($\epsilon$-approximate match and exact match). *Let $X \subseteq \mathcal{X}$ and $Y \subseteq \mathcal{Y}$ be two subsets of neurons. $\forall \epsilon \in [0, 1)$, we say $(X, Y)$ forms an $\epsilon$-approximate match in $(\mathcal{X}, \mathcal{Y})$, if*
  1. *$\forall x \in X, \mathrm{dist}(\mathbf{z}_x, \mathrm{span}(\mathbf{z}_Y)) \leq \epsilon|\mathbf{z}_x|$,*
  2. *$\forall y \in Y, \mathrm{dist}(\mathbf{z}_y, \mathrm{span}(\mathbf{z}_X)) \leq \epsilon|\mathbf{z}_y|$.*

*Here we use the $L_2$ distance: for any vector $\mathbf{z}$ and any subspace $S$, $\mathrm{dist}(\mathbf{z}, S) = \min_{\mathbf{z}' \in S} \|\mathbf{z} - \mathbf{z}'\|_2$. We call a 0-approximate match an exact match. Equivalently, $(X, Y)$ is an exact match if $\mathrm{span}(\mathbf{z}_X) = \mathrm{span}(\mathbf{z}_Y)$.*

## 3 A Theory of Neuron Activation Subspace Match

In this section, we will develop a theory which gives a complete characterization of the neuron activation subspace match problem. For two sets of neurons $\mathcal{X}, \mathcal{Y}$ in two networks, we show the structure of all the matches $(X, Y)$ in $(\mathcal{X}, \mathcal{Y})$. It turns out that every match $(X, Y)$ can be decomposed as a union of *simple* matches, where a simple match is an atomic match that cannot be decomposed any further.

Simple match is the most important concept in our theory. If there are many one-to-one simple matches (i.e. $|X| = |Y| = 1$), it implies that the two networks learn very similar representations at the neuron level. On the other hand, if all the simple matches have very large size (i.e. $|X|, |Y|$ are both large), it is reasonable to say that the two networks learn different representations, at least in details.

We will give mathematical characterization of the simple matches. This allows us to design efficient algorithms finding out the simple matches (Sec.4). The structures of exact and approximate match are somewhat different. In Section 3.1, we present the simpler case of exact match, and in Section 3.2, we describe the more general $\epsilon$-approximate match. Without being explicitly stated, when we say match, we mean $\epsilon$-approximate match.

We begin with a lemma stating that matches are closed under union.

**Lemma 2** (Union-Close Lemma). *Let $(X_1, Y_1)$ and $(X_2, Y_2)$ be two $\epsilon$-approximate matches in $(\mathcal{X}, \mathcal{Y})$. Then $(X_1 \cup X_2, Y_1 \cup Y_2)$ is still an $\epsilon$-approximate match.*

The fact that matches are closed under union implies that there exists a unique *maximum match*.

**Definition 3** (Maximum Match). *A match $(X^*, Y^*)$ in $(\mathcal{X}, \mathcal{Y})$ is the maximum match if every match $(X, Y)$ in $(\mathcal{X}, \mathcal{Y})$ satisfies $X \subseteq X^*$ and $Y \subseteq Y^*$.*

The maximum match is simply the union of all matches. In Section 4 we will develop an efficient algorithm that finds the maximum match.

Now we are ready to give a complete characterization of all the matches. First, we point out that there can be exponentially many matches. Fortunately, every match can be represented as the union of some *simple matches* defined below. The number of simple matches is polynomial for the setting of exact match given $(\mathbf{z}_x)_{x \in \mathcal{X}}$ and $(\mathbf{z}_y)_{y \in \mathcal{Y}}$ being both linearly independent, and under certain conditions for approximate match as well.

**Definition 4** (Simple Match). *A match $(\hat{X}, \hat{Y})$ in $(\mathcal{X}, \mathcal{Y})$ is a simple match if $\hat{X} \cup \hat{Y}$ is non-empty and there exist no matches $(X_i, Y_i)$ in $(\mathcal{X}, \mathcal{Y})$ such that*

*1. $\forall i, (X_i \cup Y_i) \subsetneq (\hat{X} \cup \hat{Y})$;*

*2. $\hat{X} = \bigcup_i X_i, \hat{Y} = \bigcup_i Y_i$.*

With the concept of the *simple matches*, we will show the Decomposition Theorem: every match can be decomposed as the union of a set of simple matches. Consequently, simple matches fully characterize the structure of matches.

**Theorem 5** (Decomposition Theorem). *Every match $(X, Y)$ in $(\mathcal{X}, \mathcal{Y})$ can be expressed as a union of simple matches. Formally, there are simple matches $(\hat{X}_i, \hat{Y}_i)$ satisfying $X = \bigcup_i \hat{X}_i$ and $Y = \bigcup_i \hat{Y}_i$.*

## 3.1 Structure of Exact Matches

The main goal of this and the next subsection is to understand the simple matches. The definition of simple match only tells us it cannot be decomposed. But how to find the simple matches? How many simple matches exist? We will answer these questions by giving a characterization of the simple match. Here we consider the setting of exact match, which has a much simpler structure than approximate match.

An important property for exact match is that matches are closed under intersection.

**Lemma 6** (Intersection-Close Lemma). *Assume $(\mathbf{z}_x)_{x \in \mathcal{X}}$ and $(\mathbf{z}_y)_{y \in \mathcal{Y}}$ are both linearly independent. Let $(X_1, Y_1)$ and $(X_2, Y_2)$ be exact matches in $(\mathcal{X}, \mathcal{Y})$. Then, $(X_1 \cap X_2, Y_1 \cap Y_2)$ is still an exact match.*

It turns out that in the setting of exact match, simple matches can be explicitly characterized by $v$-minimum match defined below.

**Definition 7** ($v$-Minimum Match). *Given a neuron $v \in \mathcal{X} \cup \mathcal{Y}$, we define the $v$-minimum match to be the exact match $(X_v, Y_v)$ in $(\mathcal{X}, \mathcal{Y})$ satisfying the following properties:*

*1. $v \in X_v \cup Y_v$;*

*2. any exact match $(X, Y)$ in $(\mathcal{X}, \mathcal{Y})$ with $v \in X \cup Y$ satisfies $X_v \subseteq X$ and $Y_v \subseteq Y$.*

Every neuron $v$ in the maximum match $(X^*, Y^*)$ has a unique $v$-minimum match, which is the intersection of all matches that contain $v$. For a neuron $v$ not in the maximum match, there is no $v$-minimum match because there is no match containing $v$.

The following theorem states that the simple matches are exactly $v$-minimum matches.

**Theorem 8.** *Assume $(\mathbf{z}_x)_{x \in \mathcal{X}}$ and $(\mathbf{z}_y)_{y \in \mathcal{Y}}$ are both linearly independent. Let $(X^*, Y^*)$ be the maximum (exact) match in $(\mathcal{X}, \mathcal{Y})$. $\forall v \in X^* \cup Y^*$, the $v$-minimum match is a simple match, and every simple match is a $v$-minimum match for some neuron $v \in X^* \cup Y^*$.*

Theorem 8 implies that the number of simple exact matches is at most linear with respect to the number of neurons given the activation vectors being linearly independent, because the $v$-minimum match for each neuron $v$ is unique. We will give a polynomial time algorithm in Section 4 to find out all the $v$-minimum matches.

## 3.2 Structure of Approximate Matches

The structure of $\epsilon$-approximate match is more complicated than exact match. A major difference is that in the setting of approximate matches, the intersection of two matches is not necessarily a match. As a consequence, there is no $v$-minimum match in general. Instead, we have $v$-*minimal match*.

**Definition 9** ($v$-Minimal Match). *$v$-minimal matches are matches $(X_v, Y_v)$ in $(\mathcal{X}, \mathcal{Y})$ with the following properties:*

    *1. $v \in X_v \cup Y_v$;*
    *2. if a match $(X, Y)$ with $X \subseteq X_v$ and $Y \subseteq Y_v$ satisfies $v \in X \cup Y$, then $(X, Y) = (X_v, Y_v)$.*

Different from the setting of exact match where $v$-minimum match is unique for a neuron $v$, there may be multiple $v$-minimal matches for $v$ in the setting of approximate match, and in this setting simple matches can be characterized by $v$-minimal matches instead. Again, for any neuron $v$ not in the maximum match $(X^*, Y^*)$, there is no $v$-minimal match because no match contains $v$.

**Theorem 10.** *Let $(X^*, Y^*)$ be the maximum match in $(\mathcal{X}, \mathcal{Y})$. $\forall v \in X^* \cup Y^*$, every $v$-minimal match is a simple match, and every simple match is a $v$-minimal match for some $v \in X^* \cup Y^*$.*

**Remark 1.** *We use the notion $v$-minimal match for $v \in \mathcal{X} \cup \mathcal{Y}$. That is, the neuron can be in either networks. We emphasize that this is necessary. Restricting $v \in \mathcal{X}$ (or $v \in \mathcal{Y}$) does not yield Theorem 10 anymore. In other word, $v$-minimal matches for $v \in \mathcal{X}$ do not represent all simple matches. See Remark A.1 in the Supplementary Material for details.*

**Remark 2.** *One may have the impression that the structure of match is very simple. This is not exactly the case. Here we point out the complicated aspect:*

    *1. Matches are not closed under the difference operation, even for exact matches. More generally, let $(X_1, Y_1)$ and $(X_2, Y_2)$ be two matches with $X_1 \subsetneq X_2, Y_1 \subsetneq Y_2$. $(X_2 \backslash X_1, Y_2 \backslash Y_1)$ is not necessarily a match.*
    *2. The decomposition of a match into the union of simple matches is not necessarily unique. See Section C in the Supplementary Material for details.*

## 4 Algorithms

In this section, we will give an efficient algorithm that finds the maximum match. Based on this algorithm, we further give an algorithm that finds all the simple matches, which are precisely the $v$-minimum/minimal matches as shown in the previous section. The algorithm for finding the maximum match is given in Algorithm 1. Initially, we guess the maximum match $(X^*, Y^*)$ to be $X^* = \mathcal{X}, Y^* = \mathcal{Y}$. If there is $x \in X^*$ such that $\text{dist}(\mathbf{z}_x, \text{span}(\mathbf{z}_{Y^*})) > \epsilon$, then we remove $x$ from $X^*$. Similarly, if for some $y \in Y^*$ such that $y$ cannot be linearly expressed by $\mathbf{z}_{X^*}$ within error $\epsilon$, then we remove $y$ from $Y^*$. $X^*$ and $Y^*$ are repeatedly updated in this way until no such $x, y$ can be found.

---

**Algorithm 1** max_match$((\mathbf{z}_{v'})_{v' \in \mathcal{X} \cup \mathcal{Y}}, \epsilon)$

---

1: $(X^*, Y^*) \leftarrow (\mathcal{X}, \mathcal{Y})$
2: changed $\leftarrow$ **true**
3: **while** changed **do**
4:     changed $\leftarrow$ **false**
5:     **for** $x \in X^*$ **do**
6:         **if** $\text{dist}(\mathbf{z}_x, \text{span}(\mathbf{z}_{Y^*})) > \epsilon$ **then**
7:             $X^* \leftarrow X^* \backslash \{x\}$
8:             changed $\leftarrow$ **true**
9:     **if** changed **then**
10:       changed $\leftarrow$ **false**
11:       **for** $y \in Y^*$ **do**
12:          **if** $\text{dist}(\mathbf{z}_y, \text{span}(\mathbf{z}_{X^*})) > \epsilon$ **then**
13:             $Y^* \leftarrow Y^* \backslash \{y\}$
14:             changed $\leftarrow$ **true**
15: **return** $(X^*, Y^*)$

---

**Theorem 11.** *Algorithm 1 outputs the maximum match and runs in polynomial time.*

Our next algorithm (Algorithm 2) is to output, for a given neuron $v \in \mathcal{X} \cup \mathcal{Y}$, the $v$-minimum match (for exact match given the activation vectors being linearly independent) or one $v$-minimal match (for

approximate match). The algorithm starts from $(X_v, Y_v)$ being the maximum match and iteratively finds a smaller match $(X_v, Y_v)$ keeping $v \in X_v \cup Y_v$ until further reducing the size of $(X_v, Y_v)$ would have to violate $v \in X_v \cup Y_v$.

---

**Algorithm 2** min_match$((\mathbf{z}_{v'})_{v' \in \mathcal{X} \cup \mathcal{Y}}, v, \epsilon)$

---

1: $(X_v, Y_v) \leftarrow$ max_match$((\mathbf{z}_{v'})_{v' \in \mathcal{X} \cup \mathcal{Y}}, \epsilon)$
2: **if** $v \notin X_v \cup Y_v$ **then**
3:     **return** "failure"
4: **while** there exists $u \in X_v \cup Y_v$ unchecked **do**
5:     Pick an unchecked $u \in X_v \cup Y_v$ and mark it as checked
6:     **if** $u \in X_v$ **then**
7:         $(X, Y) \leftarrow (X_v \backslash \{u\}, Y_v)$
8:     **else**
9:         $(X, Y) \leftarrow (X_v, Y_v \backslash \{u\})$
10:     $(X^*, Y^*) \leftarrow$ max_match$((\mathbf{z}_{v'})_{v' \in X \cup Y}, \epsilon)$
11:     **if** $v \in (X^*, Y^*)$ **then**
12:         $(X_v, Y_v) \leftarrow (X^*, Y^*)$
13: **return** $(X_v, Y_v)$

---

**Theorem 12.** *Algorithm 2 outputs one $v$-minimal match for the given neuron $v$. If $\epsilon = 0$ (exact match), the algorithm outputs the unique $v$-minimum match provided $(\mathbf{z}_x)_{x \in \mathcal{X}}$ and $(\mathbf{z}_y)_{y \in \mathcal{Y}}$ are both linearly independent. Moreover, the algorithm always runs in polynomial time.*

Finally, we show an algorithm (Algorithm 3) that finds all the $v$-minimal matches in time $L^{O(N_v)}$. Here, $L$ is the size of the input ($L = (|\mathcal{X}| + |\mathcal{Y}|) \cdot d$) and $N_v$ is the number of $v$-minimal matches for neuron $v$. Note that in the setting of $\epsilon = 0$ (exact match) with $(\mathbf{z}_x)_{x \in \mathcal{X}}$ and $(\mathbf{z}_y)_{y \in \mathcal{Y}}$ being both linearly independent, we have $N_v \leq 1$, so Algorithm 3 runs in polynomial time in this case.

Algorithm 3 finds all the $v$-minimal matches one by one by calling Algorithm 2 in each iteration. To make sure that we never find the same $v$-minimal match twice, we always delete a neuron in every previously-found $v$-minimal match before we start to find the next one.

---

**Algorithm 3** all_min_match$((\mathbf{z}_{v'})_{v' \in \mathcal{X} \cup \mathcal{Y}}, v, \epsilon)$

---

1: $\mathcal{S} \leftarrow \emptyset$
2: found $\leftarrow$ **true**
3: **while** found **do**
4:     found $\leftarrow$ **false**
5:     Let $\mathcal{S} = \{(X_1, Y_1), (X_2, Y_2), \cdots, (X_{|\mathcal{S}|}, Y_{|\mathcal{S}|})\}$
6:     **while** $\neg$found **and** there exists $(u_1, u_2, \cdots, u_{|\mathcal{S}|}) \in (X_1 \cup Y_1) \times (X_2 \cup Y_2) \times \cdots \times (X_{|\mathcal{S}|} \cup Y_{|\mathcal{S}|})$ unchecked **do**
7:         Pick the next unchecked $(u_1, u_2, \cdots, u_{|\mathcal{S}|}) \in (X_1 \cup Y_1) \times (X_2 \cup Y_2) \times \cdots \times (X_{|\mathcal{S}|} \cup Y_{|\mathcal{S}|})$ and mark it as checked
8:         $(X, Y) \leftarrow (\mathcal{X}, \mathcal{Y})$
9:         **for** $i = 1, 2, \cdots, |\mathcal{S}|$ **do**
10:             **if** $u_i \in \mathcal{X}$ **then**
11:                 $X \leftarrow X \backslash \{u_i\}$
12:             **else**
13:                 $Y \leftarrow Y \backslash \{u_i\}$
14:         **if** min_match$((\mathbf{z}_{v'})_{v' \in X \cup Y}, v, \epsilon)$ doesn't return "failure" **then**
15:             $(X_v, Y_v) \leftarrow$ min_match$((\mathbf{z}_{v'})_{v' \in X \cup Y}, v, \epsilon)$
16:             $\mathcal{S} \leftarrow \mathcal{S} \cup \{(X_v, Y_v)\}$
17:             found $\leftarrow$ **true**
18: **return** $\mathcal{S}$

---

**Theorem 13.** *Algorithm 3 outputs all the $N_v$ different $v$-minimal matches in time $L^{O(N_v)}$. With Algorithm 3, we can find all the simple matches by exploring all $v \in \mathcal{X} \cup \mathcal{Y}$ based on Theorem 10.*

In the worst case, Algorithm 3 is not polynomial time, as $N_v$ is not upper bounded by a constant in general. However, under assumptions we call strong linear independence and stability, we show that Algorithm 3 runs in polynomial time. Specifically, we say $(\mathbf{z}_x)_{x \in \mathcal{X}}$ satisfies $\theta$-strong linear independence for $\theta \in (0, \frac{\pi}{2}]$ if $\mathbf{0} \notin \mathbf{z}_{\mathcal{X}}$ and for any two non-empty disjoint subsets $X_1, X_2 \subseteq \mathcal{X}$, the angle between $\mathrm{span}(\mathbf{z}_{X_1})$ and $\mathrm{span}(\mathbf{z}_{X_2})$ is at least $\theta$. Here, the angle between two subspaces is defined to be the minimum angle between non-zero vectors in the two subspaces. We define $\theta$-strong linear independence for $(\mathbf{z}_y)_{y \in \mathcal{Y}}$ similarly. We say $(\mathbf{z}_x)_{x \in \mathcal{X}}$ and $(\mathbf{z}_y)_{y \in \mathcal{Y}}$ satisfy $(\epsilon, \lambda)$-stability for $\epsilon \geq 0$ and $\lambda > 1$ if $\forall x \in \mathcal{X}, \forall Y \subseteq \mathcal{Y}, \mathrm{dist}(\mathbf{z}_x, \mathrm{span}(\mathbf{z}_Y)) \notin (\epsilon|\mathbf{z}_x|, \lambda\epsilon|\mathbf{z}_x|]$ and $\forall y \in \mathcal{Y}, \forall X \subseteq \mathcal{X}, \mathrm{dist}(\mathbf{z}_y, \mathrm{span}(\mathbf{z}_X)) \notin (\epsilon|\mathbf{z}_y|, \lambda\epsilon|\mathbf{z}_y|]$. We prove the following theorem.

**Theorem 14.** *Suppose $\exists \theta \in (0, \frac{\pi}{2}]$ such that $(\mathbf{z}_x)_{x \in \mathcal{X}}$ and $(\mathbf{z}_y)_{y \in \mathcal{Y}}$ both satisfy $\theta$-strong linear independence and $(\epsilon, \frac{2}{\sin\theta} + 1)$-stability. Then, $\forall v \in \mathcal{X} \cup \mathcal{Y}, N_v \leq 1$. As a consequence, Algorithm 3 finds all the $v$-minimal matches in polynomial time, and we can find all the simple matches in polynomial time by exploring all $v \in \mathcal{X} \cup \mathcal{Y}$ based on Theorem 10.*

# 5 Experiments

We conduct experiments on architectures of VGG[Simonyan and Zisserman, 2014] and ResNet [He et al., 2016] on the dataset CIFAR10[Krizhevsky et al.] and ImageNet[Deng et al., 2009]. Here we investigate multiple networks initialized with different random seeds, which achieve reasonable accuracies. Unless otherwise noted, we focus on the neurons activated by ReLU.

The *activation vector* $\mathbf{z}_v$ mentioned in Section 2 is defined as the activations of one neuron $v$ over the validation set. For a fully connected layer, $\mathbf{z}_v \in \mathbb{R}^d$, where $d$ is the number of images. For a convolutional layer, the activations of one neuron $v$, given the image $I_i$, is a feature map $z_v(a_i) \in \mathbb{R}^{h \times w}$. We vectorize the feature map as $vec(z_v(a_i)) \in \mathbb{R}^{h \times w}$, and thus $\mathbf{z}_v := (vec(z_v(a_1)), vec(z_v(a_2)), \cdots, vec(z_v(a_d))) \in \mathbb{R}^{h \times w \times d}$.

## 5.1 Maximum Match

We introduce *maximum matching similarity* to measure the overall similarity between sets of neurons. Given two sets of neurons $\mathcal{X}, \mathcal{Y}$ and $\epsilon$, algorithm 1 outputs the maximum match $X^*, Y^*$. The maximum matching similarity $s$ under $\epsilon$ is defined as $s(\epsilon) = \frac{|X^*| + |Y^*|}{|\mathcal{X}| + |\mathcal{Y}|}$

Here we only study neurons in the same layer of two networks with same architecture but initialized with different seeds. For a convolutional layer, we randomly sample $d$ from $h \times w \times d$ outputs to form an activation vector for several times, and average the maximal matching similarity.

**Different Architecture and Dataset** We examine several architectures on different dataset. For each experiment, five differently initialized networks are trained, and the maximal matching similarity is averaged over all the pairs of networks given $\epsilon$. The similarity values show little variance among different pairs, which indicates that this metric reveals a general property of network pairs. The detail of network structures and validation accuracies are listed in the Supplementary Section E.2.

Figure 1 shows maximal matching similarities of all the layers of different architectures under various $\epsilon$. From these results, we make the following conclusions:

1. For most of the convolutional layers, the maximum match similarity is very low. For deep neural networks, the similarity is almost zero. This is surprising, as it is widely believed that the convolutional layers are trained to extract specific patterns. However, the observation shows that different CNNs (with the same architecture) may learn different intermediate patterns.
2. Although layers close to the output sometimes exhibit high similarity, it is a simple consequence of their alignment to the output: First, the output vector of two networks must be well aligned because they both achieve high accuracy. Second, it is necessary that the layers before output are similar because if not, after a linear transformation, the output vectors will not be similar. Note that in Fig 1 (b) layers close to the output do not have similarity. This is because in this experiment the accuracy is relatively low. (See also in Supplementary materials that, for a trained and an untrained networks which have very different accuracies and therefore layers close to output do not have much similarity.)

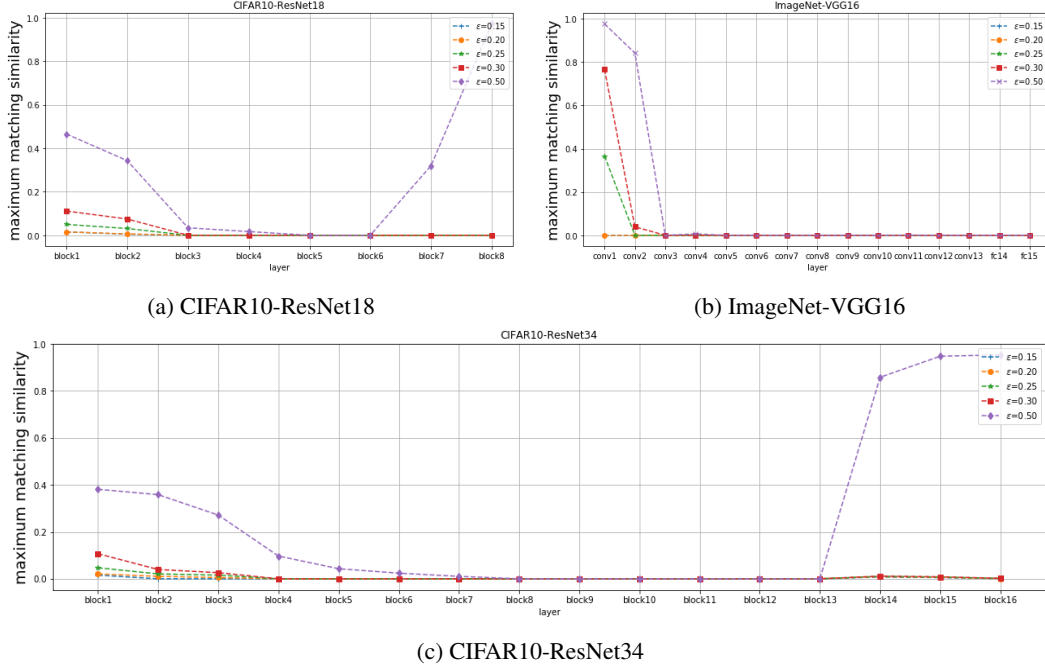

(a) CIFAR10-ResNet18                    (b) ImageNet-VGG16

(c) CIFAR10-ResNet34

Figure 1: Maximal matching similarities of different architectures on different datasets under various $\epsilon$. The x-axis is along the propagation. (a) shows ResNet18 on CIFAR10 validation set, we leave other classical architectures like VGG in Supplementary material; (b) shows VGG16 on ImageNet validation set; (c) shows a deeper ResNet on CIFAR10.

3. There is also relatively high similarity of layers close to the input. Again, this is the consequence of their alignment to the same input data as well as the low-dimension nature of the low level layers. More concretely, the fact that each low-level filter contains only a few parameters results in a low dimension space after the transformation; and it is much easier to have high similarity in low dimensional space than in high dimensional space.

## 5.2 Simple Match

The maximum matching illustrates the overall similarity but does not provide information about the relation of specific neurons. Here we analyze the distribution of the size of simple matches to reveal the finer structure of a layer. Given $\epsilon$ and two sets of neurons $\mathcal{X}$ and $\mathcal{Y}$, algorithm 3 will output all the simple matches.

For more efficient implementation, given $\epsilon$, we run the randomized algorithm 2 over each $v \in \mathcal{X} \cup \mathcal{Y}$ to get one $v$-minimal match for several iterations. The final result is the collection of all the $v$-minimal matches found (remove duplicated matches) , which we use to estimate the distribution.

Figure 2 shows the distribution of the size of simple matches on layers close to input or output respectively. We make the following observations:

1. While the layers close to output are similar overall, it seems that they do not show similarity in a local manner. There are very few simple matches with small sizes. It is also an evidence that such similarity is the result of its alignment to the output, rather than intrinsic similar representations.
2. The layer close to input shows lower similarity in the finer structure. Again, there are few simple matches with small sizes.

In sum, almost no single neuron (or a small set of neurons) learn similar representations, even in layers close to input or output.

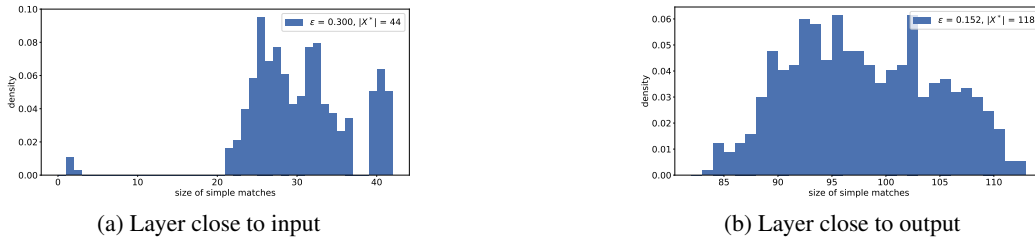

(a) Layer close to input                    (b) Layer close to output

Figure 2: The distribution of the sizes of minimal matches of layers close to input and output respectively

# 6   Conclusion

In this paper, we investigate the similarity between representations learned by two networks with identical architecture but trained from different initializations. We develop a rigorous theory and propose efficient algorithms. Finally, we apply the algorithms in experiments and find that representations learned by convolutional layers are not as similar as prevalently expected.

This raises important questions: Does our result imply two networks learn completely different representations, or subspace match is not a good metric for measuring the similarity of representations? If the former is true, we need to rethink not only learning representations, but also interpretability of deep learning. If from each initialization one learns a different representation, how can we interpret the network? If, on the other hand, subspace match is not a good metric, then what is the right metric for similarity of representations? We believe this is a fundamental problem for deep learning and worth systematic and in depth studying.

# 7   Acknowledgement

This work is supported by National Basic Research Program of China (973 Program) (grant no. 2015CB352502), NSFC (61573026) and BJNSF (L172037) and a grant from Microsoft Research Asia.

## Footnotes

[1]Li et al. [2016] also implicitly used the activation vector as the neuron's representation.

[2]In this paper we focus on neurons of the same layer. But the method applies to an arbitrary set of nerons.

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
