[Supplementary Material]

# Supplementary Material for
# Towards Understanding Learning Representations:
# To What Extent Do Different Neural Networks Learn the Same Representation

**Liwei Wang**[1,2]    **Lunjia Hu**[3]    **Jiayuan Gu**[1]    **Yue Wu**[1]
**Zhiqiang Hu**[1]    **Kun He**[4]    **John Hopcroft**[5]
[1]Key Laboratory of Machine Perception, MOE, School of EECS, Peking University
[2]Center for Data Science, Peking University, Beijing Institute of Big Data Research
[3]Computer Science Department, Stanford University
[4]Huazhong University of Science and Technology
[5]Cornell University
wanglw@cis.pku.edu.cn   lunjia@stanford.edu
{gujiayuan, frankwu, huzq}@pku.edu.cn
brooklet60@hust.edu.cn, jeh17@cornell.edu

## A    Omitted Proofs in Section 3

**Lemma 2** (Union-Close Lemma). *Let $(X_1, Y_1)$ and $(X_2, Y_2)$ be two $\epsilon$-approximate matches in $(\mathcal{X}, \mathcal{Y})$. Then $(X_1 \cup X_2, Y_1 \cup Y_2)$ is still an $\epsilon$-approximate match.*

*Proof.* The lemma follows immediately from the definition of $\epsilon$-approximate match.    $\square$

**Theorem 5** (Decomposition Theorem). *Every match $(X, Y)$ in $(\mathcal{X}, \mathcal{Y})$ can be expressed as a union of simple matches. Formally, there are simple matches $(\hat{X}_i, \hat{Y}_i)$ satisfying $X = \bigcup_i \hat{X}_i$ and $Y = \bigcup_i \hat{Y}_i$.*

*Proof.* We prove by induction on the size of the match, $|X \cup Y|$. When $|X \cup Y|$ is the smallest among all non-empty matches, we know $(X, Y)$ is itself a simple match, so the theorem holds. For larger $|X \cup Y|$, $(X, Y)$ may not be a simple match, and in this case we know $(X, Y)$ is the union of smaller matches $(X_i, Y_i)$: $X_i \cup Y_i \subsetneq X \cup Y$ and $X = \bigcup_i X_i, Y = \bigcup_i Y_i$, and thus by the induction hypothesis that every $(X_i, Y_i)$ is a union of simple matches, we know $(X, Y)$ is a union of simple matches.    $\square$

**Lemma 6** (Intersection-Close Lemma). *Assume $(\mathbf{z}_x)_{x \in \mathcal{X}}$ and $(\mathbf{z}_y)_{y \in \mathcal{Y}}$ are both linearly independent. Let $(X_1, Y_1)$ and $(X_2, Y_2)$ be exact matches in $(\mathcal{X}, \mathcal{Y})$. Then, $(X_1 \cap X_2, Y_1 \cap Y_2)$ is still an exact match.*

Lemma 6 is a direct corollary of the following claim.

**Claim A.1.** *Assume $(\mathbf{z}_x)_{x \in \mathcal{X}}$ is linearly independent. Then $\forall X_1, X_2 \subseteq \mathcal{X}, \mathrm{span}(\mathbf{z}_{X_1 \cap X_2}) = \mathrm{span}(\mathbf{z}_{X_1}) \cap \mathrm{span}(\mathbf{z}_{X_2})$.*

*Proof.* $\mathrm{span}(\mathbf{z}_{X_1 \cap X_2}) \subseteq \mathrm{span}(\mathbf{z}_{X_1}) \cap \mathrm{span}(\mathbf{z}_{X_2})$ is obvious. To show $\mathrm{span}(\mathbf{z}_{X_1}) \cap \mathrm{span}(\mathbf{z}_{X_2}) \subseteq \mathrm{span}(\mathbf{z}_{X_1 \cap X_2})$, let's consider a vector $\mathbf{z} \in \mathrm{span}(\mathbf{z}_{X_1}) \cap \mathrm{span}(\mathbf{z}_{X_2}) \subseteq \mathrm{span}(\mathbf{z}_{\mathcal{X}})$. Note that $(\mathbf{z}_x)_{x \in \mathcal{X}}$ is linearly independent, so there exists unique $\lambda_x \in \mathbb{R}$ for each $x \in \mathcal{X}$ s.t. $\mathbf{z} = \sum_{x \in \mathcal{X}} \lambda_x \mathbf{z}_x$. The

uniqueness of $\lambda_x$ and the fact that $\mathbf{z} \in \mathrm{span}(\mathbf{z}_{X_1})$ shows that $\forall x \in \mathcal{X} \backslash X_1, \lambda_x = 0$. Similarly, $\forall x \in \mathcal{X} \backslash X_2, \lambda_x = 0$. Therefore, $\lambda_x \neq 0$ only when $x \in X_1 \cap X_2$, so $\mathbf{z} \in \mathrm{span}(\mathbf{z}_{X_1 \cap X_2})$. $\qquad \square$

**Theorem 8.** *Assume $(\mathbf{z}_x)_{x \in \mathcal{X}}$ and $(\mathbf{z}_y)_{y \in \mathcal{Y}}$ are both linearly independent. Let $(X^*, Y^*)$ be the maximum (exact) match in $(\mathcal{X}, \mathcal{Y})$. $\forall v \in X^* \cup Y^*$, the $v$-minimum match is a simple match, and every simple match is the $v$-minimum match for some neuron $v \in X^* \cup Y^*$.*

*Proof.* We show that under the assumption of Theorem 8 the concept of $v$-minimum match and the concept of $v$-minimal match (Definition 9) coincide so Theorem 8 is a special case of Theorem 10.

According to Lemma 6, $\forall v \in X^* \cup Y^*$ has a unique $v$-minimum match $(X_v, Y_v)$ being the intersection of all the matches containing $v$. Therefore, for any $v$-minimal match $(X_v', Y_v')$, it holds that $X_v \subseteq X_v', Y_v \subseteq Y_v'$, and according to Definition 9 we have $(X_v, Y_v) = (X_v', Y_v')$. $\qquad \square$

**Theorem 10.** *Let $(X^*, Y^*)$ be the maximum match in $(\mathcal{X}, \mathcal{Y})$. $\forall v \in X^* \cup Y^*$, every $v$-minimal match is a simple match, and every simple match is a $v$-minimal match for some $v \in X^* \cup Y^*$.*

*Proof.* We start by showing the first half of the lemma. To prove by contradiction, let's assume that a $v$-minimal match $(X_v, Y_v)$ can be written as the union of smaller matches $(X_i, Y_i)$, i.e., $(X_i \cup Y_i) \subsetneq (X_v \cup Y_v), X_v = \bigcup_i X_i, Y_v = \bigcup_i Y_i$. In this case, one of the matches $(X_i, Y_i)$ contains $v$, which is contradictory with the definition of $v$-minimal match.

Now we show the second half of the lemma. For any neuron $v$ in a simple match $(\hat{X}, \hat{Y})$, we consider one of the smallest matches $(X_v, Y_v)$ in $(\hat{X}, \hat{Y})$ containing $v$. Here, "smallest" means that $|X_v \cup Y_v|$ is the smallest among all matches in $(\hat{X}, \hat{Y})$ containing $v$. Note that $(\hat{X}, \hat{Y})$ is itself a match containing $v$, such a smallest match $(X_v, Y_v)$ indeed exists. The fact that $(X_v, Y_v)$ is the smallest implies that it's a $v$-minimal match. Now, trivially we have $\hat{X} = \bigcup_{v \in \hat{X} \cup \hat{Y}} X_v$ and $\hat{Y} = \bigcup_{v \in \hat{X} \cup \hat{Y}} Y_v$, and since $(\hat{X}, \hat{Y})$ is a simple match, one of $(X_v, Y_v)$ has to be equal to $(\hat{X}, \hat{Y})$, which proves the second half of the lemma. $\qquad \square$

**Remark A.1.** *One important thing to note is that "$v \in X^* \cup Y^*$" in the second half of Lemma 10 cannot be replaced by "$v \in X^*$". For example, let $\mathbf{z}_{\mathcal{X}} = \{(\sqrt{\frac{1}{2}}, \sqrt{\frac{1}{2}}, 0), (\sqrt{\frac{1}{2}}, -\sqrt{\frac{1}{2}}, 0)\}$ and $\mathbf{z}_{\mathcal{Y}} = \mathbf{z}_{\mathcal{X}} \cup \{(\sqrt{1 - \epsilon^2}, 0, \epsilon)\}$. In this case, the entire match $(\mathcal{X}, \mathcal{Y})$ cannot be expressed as a union of $x$-minimal matches for $x \in \mathcal{X}$ because no $x$-minimal match contains the vector in $\mathbf{z}_{\mathcal{Y}} \backslash \mathbf{z}_{\mathcal{X}}$. However, in the case of Theorem 8 when $\epsilon = 0$ and $(\mathbf{z}_x)_{x \in \mathcal{X}}, (\mathbf{z}_y)_{y \in \mathcal{Y}}$ are both linearly independent, we can perform the replacement. That is because in this case, every match $(X, Y)$ satisfies $|X| = |Y|$, so we know $\hat{X} = \bigcup_{v \in \hat{X}} X_v$ and $\hat{Y} \supseteq \bigcup_{v \in \hat{X}} Y_v$ imply $(\hat{X}, \hat{Y}) = (\bigcup_{v \in \hat{X}} X_v, \bigcup_{v \in \hat{X}} Y_v)$.*

# B  Omitted Proofs in Section 4

**Theorem 11.** *Algorithm 1 outputs the maximum match and runs in polynomial time.*

*Proof.* Every time we delete a neuron $x$ (or $y$) from $X^*$ (or $Y^*$) at Line 7 (or Line 13), we make sure that the activation vector $\mathbf{z}_x$ (or $\mathbf{z}_y$) cannot be linearly expressed by $\mathbf{z}_{Y^*}$ (or $\mathbf{z}_{X^*}$) within error $\epsilon$, so $(X^*, Y^*)$ always contains the maximum match. On the other hand, when the algorithm terminates, we know $\forall x \in X^*$, $\mathbf{z}_x$ is linearly expressible by $\mathbf{z}_{Y^*}$ within error $\epsilon$ and $\forall y \in Y^*$, $\mathbf{z}_y$ is linearly expressible by $\mathbf{z}_{X^*}$ within error $\epsilon$, so $(X^*, Y^*)$ is a match by definition. Therefore, the output $(X^*, Y^*)$ of Algorithm 1 is a match containing the maximum match, which has to be the maximum match itself.

Before entering each iteration of the algorithm, we make sure that at least a neuron is deleted from $X^*$ or $Y^*$ in the last iteration, so there are at most $|X \cup Y|$ iterations. Therefore, the algorithm runs in polynomial time. $\qquad \square$

**Theorem 12.** *Algorithm 2 outputs one $v$-minimal match for the given neuron $v$. If $\epsilon = 0$ (exact match), the algorithm outputs the unique $v$-minimum match provided $(\mathbf{z}_x)_{x \in \mathcal{X}}$ and $(\mathbf{z}_y)_{y \in \mathcal{Y}}$ are both linearly independent. Moreover, the algorithm always runs in polynomial time.*

*Proof.* Clearly, Algorithm 2 runs in polynomial time. If there exist at least one $v$-minimal matches, then $v$ has to be in the maximum match and thus the algorithm doesn't return "failure". Therefore, the remaining is to show that the match $(X_v, Y_v)$ returned by the algorithm is indeed a $v$-minimal match.

Clearly, the first requirement of $v$-minimal match that $v \in X_v \cup Y_v$ is obviously satisfied by the algorithm. Now we prove that the second requirement is also satisfied. Consider a match $(X_0, Y_0)$ with $X_0 \subseteq X_v$ and $Y_0 \subseteq Y_v$ that satisfies $v \in X_0 \cup Y_0$. We want to show that $(X_0, Y_0) = (X_v, Y_v)$. To prove by contradiction, let's suppose $u \in (X_v \cup Y_v) \backslash (X_0 \cup Y_0)$. Let's consider $(X, Y)$ and $(X^*, Y^*)$ at Line 11 in the iteration when $u$ is being picked by the algorithm at Line 5. Since $u \notin X_0 \cup Y_0$, we know $X_0 \subseteq X_v \backslash \{u\} \subseteq X$ and $Y_0 \subseteq Y_v \backslash \{u\} \subseteq Y$. In other words, $(X_0, Y_0)$ is a match in $(X, Y)$. Moreover, since $u \in X_v \cup Y_v$, we know the "if" condition at Line 11 is not satisfied, i.e., $v \notin (X^*, Y^*)$. Therefore, $(X_0 \cup X^*, Y_0 \cup Y^*)$ is a match in $(X, Y)$ that is strictly larger than $(X^*, Y^*)$ (note that $v \in X_0 \cup Y_0$), a contradiction with $(X^*, Y^*)$ being the maximum match in $(X, Y)$. $\qquad\square$

**Theorem 13.** *Algorithm 3 outputs all the $N_v$ different $v$-minimal matches in time $L^{O(N_v)}$. With Algorithm 3, we can find all the simple matches by exploring all $v \in \mathcal{X} \cup \mathcal{Y}$ based on Theorem 10.*

*Proof.* The fact that we remove all $u_i$ from $X$ and $Y$ in each iteration implies that every time we put a match into $\mathcal{S}$, the match is different from the existing matches in $\mathcal{S}$. Moreover, every time we put a match into $\mathcal{S}$, the match is a $v$-minimal match, so $|\mathcal{S}| \le N_v$ during the whole algorithm. Therefore, the running time of the algorithm is $L^{O(N_v)}$. The remaining is to show that $\mathcal{S}$ returned by the algorithm contains all the $v$-minimal matches. To prove by contradiction, suppose $\mathcal{S} = \{(X_1, Y_1), (X_2, Y_2), \cdots, (X_k, Y_k)\}$ while there exists a $v$-minimal match $(X_{k+1}, Y_{k+1})$ that is not in $S$. By the fact that $(X_{k+1}, Y_{k+1})$ is minimal, we know $X_i \cup Y_i$ is not a subset of $X_{k+1} \cup Y_{k+1}$ for $i = 1, 2, \cdots, k$. Therefore, there exists $u_i \in (X_i \cup Y_i) \backslash (X_{k+1} \cup Y_{k+1})$ for $i = 1, 2, \cdots, k$. Consider the iteration when we pick $(u_1, u_2, \cdots, u_k)$ at Line 7. We know the "if" condition at Line 14 is satisfied because of $(X_{k+1}, Y_{k+1})$, which then implies that $(X_{k+1}, Y_{k+1}) \in \mathcal{S}$, a contradiction. $\quad\square$

**Theorem 14.** *Suppose $\exists \theta \in (0, \frac{\pi}{2}]$ such that $(\mathbf{z}_x)_{x \in \mathcal{X}}$ and $(\mathbf{z}_y)_{y \in \mathcal{Y}}$ both satisfy $\theta$-strong linear independence and $(\epsilon, \frac{2}{\sin \theta} + 1)$-stability. Then, $\forall v \in \mathcal{X} \cup \mathcal{Y}, N_v \le 1$. As a consequence, Algorithm 3 finds all the $v$-minimal matches in polynomial time, and we can find all the simple matches in polynomial time by exploring all $v \in \mathcal{X} \cup \mathcal{Y}$ based on Theorem 10.*

Theorem 14 is a direct corollary of the following lemma.

**Lemma B.1.** *Suppose $(\mathbf{z}_x)_{x \in \mathcal{X}}$ and $(\mathbf{z}_y)_{y \in \mathcal{Y}}$ both satisfy $\theta$-strong linear independence and $(\epsilon, \frac{2}{\sin \theta} + 1)$-stability. Let $(X_1, Y_1), (X_2, Y_2)$ be two matches in $(\mathcal{X}, \mathcal{Y})$. Then, $(X_1 \cap X_2, Y_1 \cap Y_2)$ is also a match.*

*Proof.* $\forall x \in X_1 \cap X_2$, let $|\mathbf{z}_x - \sum_{y \in Y_1} \mu_y \mathbf{z}_y| \le \epsilon |\mathbf{z}_x|$ and $|\mathbf{z}_x - \sum_{y \in Y_2} \mu'_y \mathbf{z}_y| \le \epsilon |\mathbf{z}_x|$. Therefore, $|\sum_{y \in Y_1 \backslash Y_2} \mu_y \mathbf{z}_y + (\sum_{y \in Y_1 \cap Y_2} (\mu_y - \mu'_y) \mathbf{z}_y - \sum_{y \in Y_2 \backslash Y_1} \mu'_y \mathbf{z}_y)| \le 2\epsilon |\mathbf{z}_x|$, which implies that $\mathrm{dist}(\sum_{y \in Y_1 \backslash Y_2} \mu_y \mathbf{z}_y, \mathrm{span}(\mathbf{z}_{Y_2})) \le 2\epsilon |\mathbf{z}_x|$. Note that by $\theta$-strong linear independence, we have the angle between $\mathrm{span}(\mathbf{z}_{Y_1 \backslash Y_2})$ and $\mathrm{span}(\mathbf{z}_{Y_2})$ is at least $\theta$. Therefore, $|\sum_{y \in Y_1 \backslash Y_2} \mu_y \mathbf{z}_y| \sin \theta \le 2\epsilon |\mathbf{z}_x|$, i.e., $|\sum_{y \in Y_1 \backslash Y_2} \mu_y \mathbf{z}_y| \le \frac{2\epsilon |\mathbf{z}_x|}{\sin \theta}$. Together with $|\mathbf{z}_x - \sum_{y \in Y_1} \mu_y \mathbf{z}_y| \le \epsilon |\mathbf{z}_x|$, we know $|\mathbf{z}_x - \sum_{y \in Y_1 \cap Y_2} \mu_y \mathbf{z}_y| \le \epsilon |\mathbf{z}_x| + \frac{2\epsilon |\mathbf{z}_x|}{\sin \theta} = (\frac{2}{\sin \theta} + 1) \epsilon |\mathbf{z}_x|$. By $(\epsilon, \frac{2}{\sin \theta} + 1)$-stability, we know $\mathrm{dist}(\mathbf{z}_x, \mathrm{span}(\mathbf{z}_{Y_1 \cap Y_2})) \le \epsilon |\mathbf{z}_x|$. Similarly, we can prove $\mathrm{dist}(\mathbf{z}_y, \mathrm{span}(\mathbf{z}_{X_1 \cap X_2})) \le \epsilon |\mathbf{z}_y|$ for every $y \in Y_1 \cap Y_2$. $\quad\square$

## C  Complicated Aspects of the Structure of Matches

**Matches are not closed under the difference operation.** Let's consider the case where $d = 2, \mathcal{X} = \{x_1, x_2\}, \mathcal{Y} = \{y_1, y_2\}, z_{x_1} = z_{y_1} = (1, 0), z_{x_2} = (0, 1), z_{y_2} = (1, 1)$. When $\epsilon$ is sufficiently small, there are two non-empty matches in total: $(\{x_1\}, \{y_1\})$ and $(\{x_1, x_2\}, \{y_1, y_2\})$. The difference of the two matches, $(\{x_2\}, \{y_2\})$ is not a match.

**A simple match might be a proper subset of another simple match.** In the example given in the last paragraph, $(\{x_1\}, \{y_1\})$ and $(\{x_1, x_2\}, \{y_1, y_2\})$ are both simple matches, while $\{x_1\} \subsetneq \{x_1, x_2\}$ and $\{y_1\} \subsetneq \{y_1, y_2\}$.

**The decomposition of a match into a union of simple matches might not be unique.** A trivial example is a simple match $(\hat{X}_1, \hat{Y}_1)$ being a proper subset of another simple match $(\hat{X}_2, \hat{Y}_2)$, where $(\hat{X}_2, \hat{Y}_2)$ can also be decomposed as the union of $(\hat{X}_1, \hat{Y}_1)$ and $(\hat{X}_2, \hat{Y}_2)$, a different decomposition from the natural decomposition using only $(\hat{X}_2, \hat{Y}_2)$. A more non-trivial example is when $\mathcal{X} = \{x_1, x_2, x_3\}, \mathcal{Y} = \{y_1, y_2, y_3\}, \mathbf{z}_{x_1} = (0, 1), \mathbf{z}_{y_1} = (1, 0), \mathbf{z}_{x_2} = \mathbf{z}_{y_2} = (1, 1), \mathbf{z}_{x_3} = \mathbf{z}_{y_3} = (1, -1)$. When $\epsilon$ is sufficiently small, the entire match $(\mathcal{X}, \mathcal{Y})$ can be decomposed as the union of simple matches in two different ways: the union of $(\{x_1, x_2\}, \{y_1, y_2\})$ and $(\{x_3\}, \{y_3\})$, or the union of $(\{x_1, x_3\}, \{y_1, y_3\})$ and $(\{x_2\}, \{y_2\})$.

# D   Instances without Strong Linear Independence or Stability

In this section, we prove Lemmas D.1 and D.2 to show that neither of the two assumptions in Theorem 14 can be removed. Specifically, Lemma D.1 shows that we cannot remove the strong linear independence assumption, even for $\epsilon = 0$ where the stability assumption is trivial. Lemma D.2 shows that we cannot remove the stability assumption, even when the instance satisfies $\frac{\pi}{2}$-strong linear independence (orthogonality).

**Lemma D.1.** *There exist instances $(\mathbf{z}_v)_{v \in \mathcal{X} \cup \mathcal{Y}}$ where there are more than polynomial number of simple exact matches. Moreover, the number of linear subspaces formed by these simple exact matches is also more than polynomial.*

*Proof.* Suppose $k, c \geq 2$ are positive integers and $d > \binom{ck}{k}$ is the dimension of the space $\mathbb{R}^d$ we are considering. Suppose we have a set $S$ of $d - \binom{ck}{k} + ck$ subspaces of dimension $d - 1$ in general position. In other words, the normal vectors $\mathbf{n}_s$ for all $s \in S$ are linearly independent.[1] Let $\mathcal{X} = \{x_1, x_2, \cdots, x_N\}$ and $\mathcal{Y} = \{y_1, y_2, \cdots, y_N\}$ for $N = \binom{d - \binom{ck}{k} + ck}{d - \binom{ck}{k} + (c-1)k}$. For each $\left(d - \binom{ck}{k} + (c-1)k\right)$-sized subset $S_i \subseteq S$, we independently pick two random unit vectors $\mathbf{z}_{x_i}, \mathbf{z}_{y_i}$ in the subspace $t_i := \bigcap_{s \in S_i} s$, i.e., $\mathbf{z}_{x_i}, \mathbf{z}_{y_i}$ are independently picked uniformly from $B_0(1) \cap t_i$.

Here, the size of $\mathcal{X}$ and $\mathcal{Y}$ are both $\binom{d - \binom{ck}{k} + ck}{d - \binom{ck}{k} + (c-1)k}$, which is roughly $d^k$ when $d$ is very large. We are going to show that, almost surely, there are roughly $d^{ck}$ simple matches in this case. Note that $c$ and $k$ are arbitrary at the very beginning and $d$ can grow arbitrarily large for any fixed $c$ and $k$, this leads to the correctness of Lemma D.1.

First, for any $S' \subseteq S$ with size $|S'| > d - \binom{ck}{k}$, we show that the number of neurons $x \in \mathcal{X}$ with $\mathbf{z}_x \in h := \bigcap_{s \in S'} s$ is almost surely less than $d - |S'|$, and by symmetry, this claim also holds for neurons $y \in \mathcal{Y}$. The claim is obvious for $|S'| > d - \binom{ck}{k} + (c-1)k$, because in this case, almost surely, there isn't any neuron $x \in \mathcal{X}$ with $\mathbf{z}_x \in h$, while $d - |S'| \geq d - |S| = \binom{ck}{k} - ck > 0$. Now we consider the case where $d - \binom{ck}{k} < |S'| \leq d - \binom{ck}{k} + (c-1)k$. Let $\ell := |S'| - (d - \binom{ck}{k}) \in \{1, 2, \cdots, (c-1)k\}$. According to our procedure, $S_i$ are $d - \binom{ck}{k} + (c-1)k$ sized subsets of $S$, so there are $\binom{ck - \ell}{k}$ different $S_i$ containing $S'$. Therefore, almost surely, the number of neurons $x \in X$ with $\mathbf{z}_x \in h$ is exactly $\binom{ck - \ell}{k}$. The rest is to show that $\binom{ck - \ell}{k} < d - |S'|$. In fact,

$$d - |S'| - \binom{ck - \ell}{k} = -\ell + \binom{ck}{k} - \binom{ck - \ell}{k} = -\ell + \sum_{i=0}^{\ell - 1} \binom{ck - \ell + i}{k - 1} > -\ell + \sum_{i=0}^{\ell - 1} 1 = 0.$$ The last inequality is based on the fact that $0 < k - 1 < ck - \ell$.

The claim we showed above implies that for any $S' \subseteq S$ with size $|S'| > d - \binom{ck}{k}$, almost surely, $h := \bigcap_{s \in S'} s$ is not the subspace formed by any match, because there are not enough vectors in $h$ to span the $d - |S'|$ dimensional space $h$.

Next, we show that, almost surely, there is no match spanning a linear subspace of dimension $0 < \ell < \binom{ck}{k}$. Otherwise, by permuting the indices and considering only linearly independent vectors in a match, we can assume without loss of generality that with non-zero probability, $(\{x_1, x_2, \cdots, x_\ell\}, \{y_{\sigma(1)}, y_{\sigma(2)}, \cdots, y_{\sigma(\ell)}\})$ is a match spanning an $\ell$ dimensional space. In our procedure, $\mathbf{z}_{x_i}$ is a unit vector randomly picked from the space $t_i = \bigcap_{s \in S_i} s$ where $S_i$ is a subset of $S$. Note that if $t_i$ is not a subspace of $\hat{Y} := \mathrm{span}(\{y_{\sigma(1)}, y_{\sigma(2)}, \cdots, y_{\sigma(\ell)}\})$, then almost surely, $\mathbf{z}_{x_i}$ doesn't belong to $\hat{Y}$. Therefore, ignoring the event of probability zero, we know that with non-zero probability every $t_i$ is a subspace of $\hat{Y}$ for $i = 1, 2, \cdots, \ell$, i.e., $\sum_{i=1}^{\ell} t_i \subseteq \hat{Y}$. Here, the summation is over linear subspaces and the sum of linear subspaces is defined to be the linear space spanned by the subspaces. Using the fact that $A^\perp + B^\perp = (A \cap B)^\perp$, we have $t_i = \bigcap_{s \in S_i}(s^\perp)^\perp = (\sum_{s \in S_i} s^\perp)^\perp$, so $\sum_{i=1}^{\ell} t_i = (\bigcap_{i=1}^{\ell} \sum_{s \in S_i} s^\perp)^\perp = (\sum_{s \in \bigcap_{i=1}^{\ell} S_i} s^\perp)^\perp$. The correctness of the last equality is because $s^\perp = \mathrm{span}(\{\mathbf{n}_s\})$ are linearly independent for different $s$ (see Claim A.1). Therefore, $\sum_{i=1}^{\ell} t_i = \bigcap_{s \in \bigcap_{i=1}^{\ell} S_i} s$, so the dimension of $\sum_{i=1}^{\ell} t_i$ is $d - |\bigcap_{i=1}^{\ell} S_i|$ while the dimension of $\hat{Y}$ is $\ell$, and thus $d - |\bigcap_{i=1}^{\ell} S_i| \le \ell < \binom{ck}{k}$. According to the claim we showed before, the number of neurons in $X$ with $\mathbf{z}_x$ belonging to $\sum_{i=1}^{\ell} t_i$ is almost surely less than $d - |\bigcap_{i=1}^{\ell} S_i| \le \ell$, which is a contradiction.

Then, we show that, almost surely, $(\mathbf{z}_x)_{x \in X}$ for $X \subseteq \mathcal{X}$ with size $|X| \le \binom{ck}{k}$ is linearly independent, and by symmetry this also holds for $Y \subseteq \mathcal{Y}$. Otherwise, the smallest subset $X$ making $(\mathbf{z}_x)_{x \in X}$ not linearly independent with non-zero probability has size $\ell \le \binom{ck}{k}$. Without loss of generality, we assume $X = \{x_1, x_2, \cdots, x_\ell\}$. $X$ has the minimum size implies that, almost surely conditioned on $(\mathbf{z}_x)_{x \in X}$ being not linearly independent, $\mathbf{z}_{x_i} \in \mathrm{span}(\{\mathbf{z}_x : x \in X \setminus \{x_i\}\}) = \mathrm{span}(\{\mathbf{z}_x : x \in X\})$ for every $1 \le i \le \ell$. According to our procedure, $\mathbf{z}_{x_i}$ is randomly picked from $t_i$, and as before, we know that every $t_i$ is a subspace of $\mathrm{span}(\{\mathbf{z}_x : x \in X \setminus \{x_i\}\}) = \mathrm{span}(\{\mathbf{z}_x : x \in X\})$ for $i = 1, 2, \cdots, \ell$ with non-zero probability. Therefore, we know with non-zero probability, $\sum_{i=1}^{\ell} t_i \subseteq \mathrm{span}(\{\mathbf{z}_x : x \in X\})$ has dimension at most $\ell - 1 < \binom{ck}{k}$, which leads to the same contradiction as before.

Finally, for any $S' \subseteq S$ with size $|S'| = d - \binom{ck}{k}$, we show that, almost surely, $h := \bigcap_{s \in S'} s$ is the subspace spanned by a match. In this case, the dimension of $h$ is $\binom{ck}{k}$ and there are $\binom{ck}{k}$ vectors in $(v_x)_{x \in \mathcal{X}}$ (and in $(v_y)_{y \in \mathcal{Y}}$) belonging to $h$ according to our procedure. These vectors are almost surely linearly independent as we have shown before, so they span $h$, forming a match.

As we showed above, almost surely, every $S' \subseteq S$ with size $|S'| = d - \binom{ck}{k}$ is a simple match, so there are $\binom{d - \binom{ck}{k} + ck}{d - \binom{ck}{k}}$ simple matches, which is roughly $d^{ck}$ when $d$ is very large. $\square$

The following lemma shows that we cannot remove the stability assumption, even when the instance satisfies $\frac{\pi}{2}$-strong linear independence (orthogonality).

**Lemma D.2.** $\forall \epsilon \in (0, \frac{1}{3})$, there exist instances $(\mathbf{z}_v)_{v \in \mathcal{X} \cup \mathcal{Y}}$ satisfying $\frac{\pi}{2}$-strong linear independence with exponential number of simple $\epsilon$-approximate matches.

*Proof.* Let $\mathcal{X} = \{x_1, x_2, \cdots, x_n\}$ and $\mathcal{Y} = \{y_1, y_2, \cdots, y_n\}$. Suppose the dimension $d$ is equal to $n$. (If we want $d > n$, we can append zeros to the coordinate of every vector.) Let $\mathbf{z}_{x_i} =$

$(\underbrace{0, 0, \cdots, 0}_{i-1}, 1, \underbrace{0, 0, \cdots, 0}_{n-i}) \in \mathbb{R}^n$. Before we construct $\mathbf{z}_{y_i}$, we first consider a sequence of matrices $A_{2^0}, A_{2^1}, A_{2^2}, \cdots$ defined in the following way:

1. $A_1 = [1]$;

2. $A_{2m} = \begin{bmatrix} A_m & A_m \\ -A_m^T & A_m^T \end{bmatrix}$.

We choose $n$ to be a power of 2. It's easy to show by induction that $A_n A_n^T = A_n^T A_n = nI$, $A_n + A_n^T = 2I$ and every element on the diagonal of $A_n$ is 1. Now we define $\mathbf{w}_i \in \mathbb{R}^n$ to be the vector whose coordinate is the $i$th column of $A_n$ for $i = 1, 2, \cdots, n$. We have the following:

1. $|\mathbf{w}_i| = \sqrt{n}$;

2. $\mathbf{w}_i \cdot \mathbf{w}_j = 0$ for $i \neq j$;

3. $\mathbf{w}_i \mathbf{z}_{x_i} = 1$;

4. $\mathbf{w}_i \mathbf{z}_{x_j} = \pm 1$;

5. $\mathbf{w}_i \mathbf{z}_{x_j} + \mathbf{w}_j \mathbf{z}_{x_i} = 0$ for $i \neq j$.

Let $\delta = \sqrt{\frac{2\epsilon^2}{n}}$. We define $\mathbf{z}_{y_i} = (\sqrt{1 - (n-1)\delta^2} - \delta)\mathbf{z}_{x_i} + \delta \mathbf{w}_i$. Now we have the following:

1. $|\mathbf{z}_{y_i}| = 1$;

2. $\mathbf{z}_{y_i} \cdot \mathbf{z}_{y_j} = 0$ for $i \neq j$;

3. $\mathbf{z}_{x_i} \cdot \mathbf{z}_{y_i} = \sqrt{1 - (n-1)\delta^2}$;

4. $\mathbf{z}_{x_i} \cdot \mathbf{z}_{y_j} = \pm \delta$ for $i \neq j$.

Now, let's consider an $\epsilon$-approximate match $(X, Y)$ in $(\mathcal{X}, \mathcal{Y})$. Suppose $y_i \in Y$. We show by contradiction that $x_i \in X$. Suppose $x_i \notin X$. Then $\text{dist}(\mathbf{z}_{y_i}, \text{span}(\mathbf{z}_X)) = \sqrt{1 - |X|\delta^2} \geq \sqrt{1 - n\delta^2} = \sqrt{1 - 2\epsilon^2} > \epsilon$, which is a contradiction. Therefore, as long as $y_i \in Y$, we know $x_i \in X$. For the same reason, as long as $x_i \in X$, we know $y_i \in Y$. Therefore, $\exists S \subseteq \{1, 2, \cdots, n\}$ s.t. $X = \{x_i : i \in S\}, Y = \{y_i : i \in S\}$.

Now we show that $(\{x_i : i \in S\}, \{y_i : i \in S\})$ is an $\epsilon$-approximate match if and only if $|S| \geq \frac{n}{2}$. Actually, we have $\forall j \in S, \text{dist}(\mathbf{z}_{x_j}, \text{span}(\{\mathbf{z}_{y_i} : i \in S\})) = \text{dist}(\mathbf{z}_{y_j}, \text{span}(\{\mathbf{z}_{x_i} : i \in S\})) = \sqrt{(n - |S|)\delta^2} = \sqrt{\frac{2(n-|S|)}{n}}\epsilon$, and we know $\sqrt{\frac{2(n-|S|)}{n}}\epsilon \leq \epsilon$ if and only if $|S| \geq \frac{n}{2}$. Therefore, the number of simple matches is $\binom{n}{\lceil \frac{n}{2} \rceil}$, which is exponential in $n$. $\square$

# E  Additional Experiment Results

## E.1  Different Architectures

Besides ResNet18, VGG16 and ResNet34, we also train differently initialized neural networks like VGG11 and VGG13. Figure 1 shows the maximum matching similarities of all the layers of VGG13 and ResNet10. The result is similar to what is mentioned in Section 5, which implies that our conclusions might apply on most modern deep networks.

(a) CIFAR10-VGG11

(b) CIFAR10-VGG13

(c) CIFAR10-VGG16

Figure 1: Maximum matching similarities of all the layers of different architectures under various $\epsilon$.

(a) Two untrained networks using the same distribution for random initialization

(b) Untrained vs. Fine-Trained

Figure 2: Maximum match similarity between networks at different stages. Figure(a) shows the similarity of two untrained network. Figure(b) shows the similarity of the same network at different stages.

## E.2    Details of Architecture

| VGG | stage1(64) | stage2(128) | stage3(256) | stage4(512) | stage5(512) | accuracy |
|---|---|---|---|---|---|---|
| vgg11 | conv$\times$1 | conv$\times$1 | conv$\times$1 | conv$\times$1 | conv$\times$1 | 91.10% |
| vgg13 | conv$\times$2 | conv$\times$2 | conv$\times$2 | conv$\times$2 | conv$\times$2 | 92.78% |
| vgg16 | conv$\times$2 | conv$\times$2 | conv$\times$3 | conv$\times$3 | conv$\times$3 | 92.84% |
| ResNet | stage1(64) | stage2(128) | stage3(256) | stage4(512) | | accuracy |
| resnet18 | block$\times$2 | block$\times$2 | block$\times$2 | block$\times$2 | | 94.24% |
| resnet34 | block$\times$3 | block$\times$4 | block$\times$6 | block$\times$3 | | 95.33% |

Table 1: Structure of architecture (fully connected layers are omitted) and validation accuracy

Figure 3: Visualization of neurons in *fc2*. Each row includes top 9 images that maximize the activation of one neuron. The neurons in the same network are illustrated on the same side.

## E.3    Max Match during Training

As can be seen in Figure 2, after initialization, two untrained networks show similarity to some extent, while the untrained network and the trained network are much more different with each other. We believe the similarity between two untrained networks is due to the fact that the initialized networks take all its parameters from the same distributions like Xaiver initialization. Since we have many neurons that can be seen as drawn from the same distribution, we may observe phenomenon like this. Meanwhile, after training, the network turns to be quite different.

## E.4    Neuron Visualization

Our experiments show that there exist a few simple matches of small size. We randomly choose a pair of networks to produce two simple matches for two fully connected layers and visualize them following the common practice. Figure 3 and 4 visualize top 9 images that maximize the activation of each neuron in simple matches.

Figure 4: Visualization of neurons in *fc1*. Each row includes top 9 images that maximize the activation of one neuron. The neurons in the same network are illustrated on the same side.

## Footnotes

[1] Note that when $k, c \geq 2$, we have $d - \binom{ck}{k} + ck < d$.