[Reviews · NeurIPS 2018]

Reviewer 1



This work attempts to provide a theory/definition for how to define a match between two clusters of neurons (e.g. two layers), each from a neural network. The definition (Definition 1, in paper) allows one to study the similarity between a pair of neural clusters (of the same size). In this paper, this definition of match is employed in this paper to study two networks of the same architecture but trained with different random initializations. This work seems to be a follow-up work from Convergent Learning [1] and SVCCA [2]. The authors characterize the representation of a neural cluster C as the subspace spanned by the set of activation vectors of C. This representation is based on the general assumption that neural activations in a layer are followed by affine transformations and studied neural clusters with ReLU activation functions. An algorithm for finding the maximum match between two neural groups are also provided. The empirical results are tested on a few VGG variants trained on CIFAR-10. + Clarity - The paper is very well-written and clearly presented. - Notations are defined properly. Intuition provided for the definitions. + Significance - This work provides an important tool for quantifying the similarity between two neural clusters, and might have an impact to the community. + Quality - Overall, this is a solid work with a strong theoretically motivated characterization of similarity of two neural clusters. - The empirical results could be stronger to better support the theory provided. - It would be great if the authors could provide a comparison or connection between their match definition with measuring the mutual information among two neural groups. - I would love to see an experiment on quantifying the similarity between layers in a base network and a transferred network in transfer learning setup where the base and target tasks are similar. Would we see expected results as in Yosinski et al. 2015 [3]? + Originality - Novel method. Overall, an interesting paper! A clear accept! [1] Li, Yixuan, et al. "Convergent learning: Do different neural networks learn the same representations?." ICLR. 2016. [2] Raghu, Maithra, et al. "Svcca: Singular vector canonical correlation analysis for deep learning dynamics and interpretability." NIPS. 2017. [3] Yosinski, Jason, et al. "How transferable are features in deep neural networks?." Advances in neural information processing systems. 2014. ---- After rebuttal ----- I have read all the reviews and the rebuttal. My final recommendation stays the same: accept! Great work!

Reviewer 2



This works explores a novel matching procedure between activations in neural networks, that can be used to determine whether two identical networks initialized from a different seed converge to the same intermediate representations. With a carful laid theory, a suitable and provably efficient algorithms and empirical experiments, the authors give non-trivial insights into the representations learned by networks. The authors establish in an eloquent and concise way the necessary concepts for their exact and approximately matching activations metric. I enjoyed both the convincing motivation given and the carful treatment all the way to a practical algorithm that can be easily used. I was, however, a bit disappointed from the empirical experiments. I feel that this work could benefit from a wider range of networks tested and variety of activations. By doing a more comprehensive comparison we might be able to learn how different architectures behave and get more practical insights which a currently a bit lacking to my taste. Strengths - A novel approach with thorough survey on past attempts - Good theoretical and justifiable argument, with efficient algorithms - Interesting results regarding convolutional vs fully-connected layers Weakness - Empirical results could benefit from a wider range of architectures and tasks (currently only VGG on cifar10). Some could argue that they are currently not convincing. Edit: Following rebuttal, I've decided to update my review and recommend acceptance. I recommend that additional empirical results be referred to in main text.

Reviewer 3



Overall I liked the method developed by this paper, it’s an interesting alternative to the SVCCA method of [1]. The paper is very well written, and displays a high level of mathematical rigor. My main disappointment is the authors do not do much with their new technique, the main experimental observation is the middle layers of two independently trained VGG networks are not similar compared to the bottom and top layers. This is intriguing and I’m really curious as to why, do the authors have any insight into this? I’d also be curious to interpret what very small simple matches represent. One could for example pick the max variance linear combination of neurons in a simple match and visualize the validation images by projection on this subspace, if the concept of a simple match were meaningful then we might be able to interpret this by visualizing the images sorted along these projections. I’d also like to see experiments regarding a sort of “null hypothesis” for this sort of analysis. For example, what happens if we run these similarity measures to compare a trained network to an untrained network, maybe we would still see matches just by spurious correlations. Did the authors investigate this? How large does the maximum match need to be before we should conclude the layers are similar? Also how does this method account for the possibility of completely distributed representations? That is the method looks for subsets of neurons between two networks that have the same span. However, one could imagine a setting where there is a particular linear combination of all neurons in X that have the exact same span of a different linear combination of all neurons in Y, but no strict subset of X and Y match according to the definition in this work. In this case one it seems reasonable to conclude that these two linear combinations compute the same representation (at least assuming the match holds on enough data points), but the proposed method might not pick this up if no strict subset of X and Y have the same span. Can the authors comment on this possibility? A few typos, and other comments: 31 “understand” 267 “neural networks” 272: The conclusion that the lower layers are not similar based on the fact that high layer similarity only occurs at epsilon=0.5 seems a bit heuristic to me. How do we measure similarity of layers as a function of epsilon and layer similarity? Can the authors give some insight as to how to interpret the size of epsilon and maximum matching similarity? https://arxiv.org/abs/1706.05806 Edit: In light of the rebuttal I'm changing my score to 7. I did not see all of the additional experiments in the Supplementary material on my reading, and found the plots visualizing the simple matches between independently trained networks to be compelling. Good work!